



# Saturation vapor pressure characterization of selected low-volatility organic compounds using a residence time chamber

Zijun Li[1], Noora Hyttinen[2], Miika Vainikka[1], Olli-Pekka Tikkasalo[3], Siegfried Schobesberger[1], and Taina Yli-Juuti[1]

[1]Department of Technical Physics, University of Eastern Finland, Kuopio, Finland

[2]Department of Chemistry, Nanoscience Center, University of Jyväskylä, Jyväskylä, Finland

[3]Natural Resources Institute Finland (Luke), Helsinki, Finland

*Correspondence to*: Zijun Li (zijun.li@uef.fi) and Taina Yli-Juuti (taina.yli-juuti@uef.fi)

**Abstract**

Saturation vapor pressure ($p_{sat}$) is an important thermodynamic property regulating the gas-to-particle partitioning of organic compounds in the atmosphere. Low-volatility organic compounds (LVOCs), with sufficiently low $p_{sat}$ values, primarily stay in the particle phase and contribute to aerosol formation. Accurate information on the $p_{sat}$ of LVOCs would require volatility measurements performed at temperatures relevant to atmospheric aerosol formation. Here, we present an isothermal evaporation method using a residence time chamber to measure $p_{sat}$ for dry, single-compound nanoparticles at 295 K. Our method is able to characterize organic compounds with $p_{sat}$ spanning from $10^{-8}$ to $10^{-4}$ Pa at 295 K. The compounds included four polyethylene glycols (PEG: PEG6, PEG7, PEG8 and PEG9), two monocarboxylic acids (palmitic acid and stearic acid), two dicarboxylic acids (azelaic acid and sebacic acid), two alcohols (meso-erythritol and xylitol), and di-2-ethylhexyl sebacate (DEHS). There was a good agreement between our measured $p_{sat}$ values and those reported by previous volatility studies using different measurement techniques, mostly within one order of magnitude. Additionally, quantum-chemistry-based COSMO*therm* calculations were performed to estimate the $p_{sat}$ values of the studied compounds. COSMO*therm* predicted the $p_{sat}$ value for most of the studied compounds within one order of magnitude difference between the experimental and computational estimates.

## 1. Introduction

Secondary organic aerosol (SOA) particles account for 64 % to 95 % of submicron organic aerosol particles measured at different locations from urban to remote areas (Zhang et al., 2007; Jimenez et al., 2009). Gas-phase oxidation of volatile organic compounds leads to ranges of organic vapors with different functionalities and thus, volatility. Organic vapors with sufficiently low volatility can condense onto existing particles or form new particles (Hallquist et al., 2009). Particular attention has been paid to low-volatility organic compounds (LVOCs) which significantly contribute to SOA formation (Ehn et al., 2014; Mohr et al., 2019; Schwantes et al., 2019). Under gas-particle equilibrium, nearly all ($\approx$ 100 %) LVOCs partition into the particle phase in the presence of ambient aerosol mass loadings. The volatility of an organic compound can be quantified by its saturation vapor pressure ($p_{sat}$) which is the key thermodynamic property describing the equilibrium gas-particle partitioning of the compound





(Pankow, 1994). LVOCs typically refer to organic compounds with $p_{sat}$ between $10^{-8}$ and $10^{-6}$ Pa (saturation mass

concentration of the order between $10^{-3}$ and $10^{-1}$ µg m$^{-3}$). Numerous measurement techniques have been used to

estimate the $p_{sat}$ values of organic compounds (e.g., dicarboxylic acids) (Bilde et al., 2015).

The thermal desorption method is typically achieved by the desorption of particle samples from a collection plate

at elevated temperature and the follow-up gas-phase measurements of the evaporating molecules with mass

spectrometers. Examples of this method include the thermal desorption particle beam mass spectrometry

(TDPB-MS; Chattopadhyay and Ziemann (2005)), thermal desorption proton transfer reaction mass spectrometry

(TPD-PT-RMS; Cappa et al. (2008), Holzinger et al. (2010), and Eichler et al. (2015)), atmospheric solids analysis

probe mass spectrometry (ASAP-MS; Bruns et al. (2012)), and Filter Inlet for Gases and AEROsols coupled with

time-of-flight chemical ionization mass spectrometer (FIGAERO-ToF-CIMS; Lopez-Hilfiker et al. (2014)).

Estimation of $p_{sat}$ using the thermal desorption method can be biased by either high sample mass loading (i.e., in

order of micrograms) (Huang et al., 2018) or thermal decomposition of heat-labile organic compounds

(Schobesberger et al., 2018; Yang et al., 2021).

For the method based on particle size changes, a population of monodisperse particles are placed in a non-

equilibrium condition so that their particle sizes decrease due to evaporation. Typically, a tandem differential

mobility analyzer (TDMA) is used to monitor particle size changes during evaporation with residence times from

seconds to minutes. Such a method can determine the volatilities of organic compounds with $p_{sat}$ as low as $10^{-6}$ Pa

(Bilde et al., 2003). The size-selected particles are in the size range from several tens to hundreds of nanometers,

with number concentrations from several tens to thousands per cm$^{3}$. Flow tube-TDMA (FT-TDMA) has been

deployed to probe the $p_{sat}$ from the evaporation of dry particles or aqueous droplets under ambient temperature and

relative humidity (RH) conditions (Koponen et al., 2007; Yli-Juuti et al., 2013; Emanuelsson et al., 2016).

Furthermore, working at elevated temperatures up to 573 K, volatility-TDMA (V-TDMA) has been used to probe

not only $p_{sat}$ but also enthalpy of vaporization or sublimation of organic compounds by passing dry aerosol particles

through a stainless steel thermodenuder at different controlled temperatures (Salo et al., 2010). To absorb the

evaporated gas molecules from particle samples, the thermodenuder can be either filled up with activated charcoal

(Saha et al., 2015) or followed by an activated charcoal scrubber (Salo et al., 2010). Since V-TDMA measurements

are performed with elevated temperatures, the $p_{sat}$ values derived from the measurements of thermally labile

compounds can be potentially biased by thermal decomposition, as in the thermal desorption method mentioned

above.

Furthermore, an electrodynamic balance (EDB) (Zardini and Krieger, 2009) or optical tweezer (Cai et al., 2015)

can be used to determine the $p_{sat}$ of organic compounds, in a manner similar to the TDMA method, by monitoring

the size changes of trapped single, micrometer-sized particles. A single particle typically in the size range of 2 –

20 µm is trapped in an environmental cell. As the cell is continuously flushed with clean air flow of controlled

temperature and RH, organic molecules keep evaporating from particle phase to gas phase in a non-equilibrium

condition and the particle size decreases accordingly. The $p_{sat}$ of an organic compound can be determined by the

optical measurement of particle size changes during evaporation (Zardini et al., 2006; Mitchem and Reid, 2008).

For organic molecules with $p_{sat}$ of $10^{-6}$ Pa or lower, using the single particle method would take more than 24 hours





to obtain measurable particle size changes with minimum size measurement errors (Cai et al., 2015; Krieger et al., 2018).

Different from the TDMA method, an integrated volume method (IVM) developed by Saleh et al. (2008) estimates $p_{sat}$ of organic compounds on the basis of the change in the total aerosol volume under a quasi-equilibrium condition.

In a typical measurement, dry, polydisperse particles pass through a thermodenuder with no absorbing material and reach equilibration with the surrounding air at a set temperature, thereby leading to changes in particle size and volume. The equilibrium condition is ensured by applying high aerosol mass concentrations (~ 500 μg m$^{-3}$) to saturate the gas phase, and residence times of tens of seconds to complete the evaporation kinetics, with no interaction between the investigated compounds and wall material of the thermodenuder (Saleh et al., 2008). Since

saturating the gas phase is essential in the IVM method, it might be challenging to apply such a method to investigate compounds of high volatilities even with mass loading of hundreds of μg m$^{-3}$. So far, the IVM method has been used to characterize the volatilities of atmospheric-relevant organic compounds with $p_{sat}$ (298 K) between $10^{-5}$ and $10^{-4}$ Pa (Saleh et al., 2008; Saleh et al., 2010; Babar et al., 2020).

Knudsen cells can be also utilized for $p_{sat}$ measurements of organic compounds. A macroscopic particle sample

will be first placed in the cell with operation pressure of $10^{-2}$ Pa or below and then allowed to establish an equilibrium with the gas phase. The $p_{sat}$ of an organic compound can be determined by measuring either the mass change with a mass balance over time (Da Silva et al., 2001) or effusion rate of gas-phase organic molecules from the cell with a mass spectrometer (Booth et al., 2009; Dang et al., 2019; Shelley et al., 2020). Using a mass balance as the detection method typically requires the experiments at elevated temperatures (e.g., up to 421 K) to ensure

sufficient material loss from samples with $p_{sat}$ (298 K) as low as $10^{-8}$ Pa (Goldfarb and Suuberg, 2008). The $p_{sat}$ measurements using this technique can be biased if the particle samples are contaminated by compounds of high volatilities (Bilde et al., 2015). When using a mass spectrometer that has high sensitivity as the detection method, volatility measurements can be performed at around ambient temperatures (298 – 338 K) (Booth et al., 2009). So far, such a technique has been used to study organic compounds with $p_{sat}$(298 K) down to $10^{-6}$ Pa in aerosol research

(Booth et al., 2009; Krieger et al., 2018).

To date, many empirical models have been developed to estimate $p_{sat}$ of organic compounds. The simplest models use elemental compositions (Donahue et al., 2011; Bianchi et al., 2019), while group-contribution methods require knowledge on the functional groups of organic compounds (Nannoolal et al., 2008; Pankow and Asher, 2008; Compernolle et al., 2011). For a multifunctional compound, the interaction between its functional groups is an

important factor for describing the intermolecular interaction in the condensed phase. The conductor-like screening model for real solvents (COSMO-RS) (Klamt, 1995; Klamt et al., 1998; Eckert and Klamt, 2002) has therefore recently been used to estimate the $p_{sat}$ of multifunctional organic compounds that are atmospherically relevant (Kurten et al., 2016; Kurtén et al., 2018; Hyttinen et al., 2021; Hyttinen et al., 2022; Stahn et al., 2022). However, few comparisons have been made between COSMO-RS-estimated and experimental $p_{sat}$ values of LVOCs, due to

the scarcity of reliable $p_{sat}$ measurements of relevant compounds.

Experimentally determining $p_{sat}$ of LVOCs is challenging. All the measurement techniques presented above have their pros and cons, and multiple techniques are needed to obtain a comprehensive view on the $p_{sat}$. Similarly new methods for determining $p_{sat}$ are desired, especially for measuring $p_{sat}$ of LVOCs that are particularly relevant for





atmospheric SOA formation. In this study, we present an isothermal particle evaporation method using a TDMA
setup for $p_{sat}$ characterization. Without any additional heating, the method can determine $p_{sat}$ values of LVOCs
down to $10^{-8}$ Pa by monitoring the evaporation of nanoparticles in a residence time chamber (RTC) under dry
conditions at room temperature (i.e., 295 K) within timescales of hours. No calibrant with known $p_{sat}$ is required.
We studied a set of LVOCs of which $p_{sat}$ values have been reported in literature. Moreover, quantum chemistry
calculations with COSMO-RS were performed to estimate $p_{sat}$ values of the selected LVOCs. A comparison
between experimental and calculated $p_{sat}$ values helps us evaluate the accuracy of the $p_{sat}$ calculations of the studied
LVOCs.

## 2. Methods

### 2.1. Particle evaporation measurements

In this study, a set of pure organic compounds were chosen for $p_{sat}$ characterization and used as received without
any purification. The used polyethylene glycol (PEG, Polypure AS) standards were PEG6, PEG7, PEG8, and PEG9.
The used mono- and dicarboxylic acids were palmitic acid (Sigma Aldrich, ≥ 99%), stearic acid (Sigma Aldrich,
95%), azelaic acid (Sigma Aldrich, 98%), and sebacic acid (Sigma Aldrich, 99%). The remaining chemicals were
meso-erythritol (Alfa Aesar, 99%), xylitol (Sigma Aldrich, ≥ 99%) and di-2-ethylhexyl sebacate (DEHS, Topas
GmbH, > 98%). The properties of individual organic compounds are summarized in Table 1. Acetonitrile (Fisher
Scientific, 99.8%) was used as a solvent for stock solutions in this study, with the exception of meso-erythritol and
xylitol which were dissolved using deionized water (18.2 MΩ).

**Table 1**. Properties of the organic compounds used in this study.

| Groups | Compounds | Chemical Formula | $M$ [a] [g mol$^{-1}$] | $\rho$ [b] [g cm$^{-3}$] | $\sigma$ [c] [mN m$^{-1}$] | $D_{air}$ [d] [m$^2$ s$^{-1}$] |
|---|---|---|---|---|---|---|
| Polyethylene Glycol | PEG 6 | $C_{12}H_{26}O_7$ | 282.30 | 1180 | 45 | $4.67\times10^{-6}$ |
| | PEG 7 | $C_{14}H_{30}O_8$ | 326.40 | 1206 | 45 | $4.32\times10^{-6}$ |
| | PEG 8 | $C_{16}H_{34}O_9$ | 370.40 | 1234 | 45 | $4.04\times10^{-6}$ |
| | PEG 9 | $C_{18}H_{38}O_{10}$ | 414.50 | 1257 | 45 | $3.80\times10^{-6}$ |
| Monocarboxylic acid | Palmitic acid | $C_{16}H_{32}O_2$ | 256.42 | 852 | 130 | $4.37\times10^{-6}$ |
| | Stearic acid | $C_{18}H_{36}O_2$ | 284.48 | 941 | 160 | $4.12\times10^{-6}$ |
| Dicarboxylic acid | Azelaic acid | $C_9H_{16}O_4$ | 188.22 | 1251 | 180 | $5.65\times10^{-6}$ |
| | Sebacic acid | $C_{10}H_{18}O_4$ | 202.25 | 1210 | 100 | $5.38\times10^{-6}$ |
| Alcohol | meso-Erythritol | $C_4H_{10}O_4$ | 122.12 | 1451 | 160 | $7.66\times10^{-6}$ |
| | Xylitol | $C_5H_{12}O_5$ | 152.15 | 1520 | 160 | $6.85\times10^{-6}$ |
| Ester | DEHS | $C_{26}H_{50}O_4$ | 342.30 | 912 | 32 | $3.37\times10^{-6}$ |

[a] molecular weight; [b] density, for references see Table S1 in the Supplement; [c] surface tension or energy, for
references see Table S2 in the Supplement; [d] gas-phase diffusivity of a compound in air calculated using Eq. (2).





Particle samples were generated by nebulizing solutions of organic compounds with an atomizer (ATM 226, Topas GmbH). Prior to the size selection, either a silica gel diffusion dryer or a large dilution flow of dry purified air was used to remove the used solvent (i.e., water or acetonitrile). Isothermal evaporation of the particles was measured in a setup similar to that used to study volatilities of biogenic SOA particles in previous studies (Yli-Juuti et al., 2017; Buchholz et al., 2019; Li et al., 2021; Li et al., 2023). The schematic diagram of the measurement setup is

shown in Figure S1 in the Supplement. Two parallel nanometer aerosol mobility analyzers (NanoDMA, model 3085, TSI) were used to select particles with 80-nm electrical mobility diameter. We operated the two NanoDMAs in an open-loop setting with a sample to sheath flow ratio of 1:8 with clean, dry air as the sheath flow. Such a configuration ensured the gas phase free of the studied compounds (Li and Chen, 2005). Eventually, the resultant monodisperse aerosol was fed into either bypass lines of varying lengths or into a stainless-steel RTC of either 25 L

or 100 L in volume for prolonged isothermal evaporation. This setup for particle evaporation measurements enables residence times from one second to nearly seven hours. Vapor wall losses in the bypass lines and RTCs were rapid enough to ensure negligible amounts of vapors in the gas phase (Yli-Juuti et al., 2017). The whole setup was maintained under dry conditions (RH < 5%) at 295 K. Before each isothermal evaporation experiment, the NanoDMAs, bypass tubing, and RTCs were flushed for hours with dry purified air.

Particle size changes due to isothermal evaporation were periodically measured with a scanning mobility particle sizer (SMPS, TSI; DMA 3080, CPC 3775). Under the assumption of particle sphericity, the extent of particle evaporation was quantified using the evaporation factor (EF). Independent of particle number concentration or mass loading, the EF was determined as $(D_{p,t}/D_{p,0})$. We chose the measured size with the least amount evaporation (residence time = 1s following size selection) as $D_{p,0}$ and the measured size after residence time $t$ of evaporation as

$D_{p,t}$.

### 2.2. Determination of p$_{sat}$ values

For each of the studied compound, the p$_{sat}$ value can be determined from the isothermal evaporation data (Riipinen et al., 2006; Salo et al., 2010). The particle size change due to evaporation can be expressed as

$$\frac{dD_p}{dt} = -p_{sat} \cdot \frac{4D_{i,air}M_i}{\rho_i D_p RT} \cdot \exp\left(\frac{4\sigma_i M_i}{\rho_i D_p RT}\right) \cdot \beta, \tag{1}$$

where $D_{i,air}$ is the gas-phase diffusivity of molecule $i$ in air, $M_i$ is the molecular weight, $\rho_i$ is the density, $R$ is the universal gas constant, $T$ is the temperature (i.e., 295 K), $\sigma_i$ is the surface tension or energy, and $\beta$ is a factor correcting the condensation mass flux in the transition regime ($D_p$ in between 0.02 and 3 µm).

The gas-phase diffusivity of molecule $i$ in air, $D_{i,air}$, can be estimated using Fuller's semi-empirical method (Fuller et al., 1966)

$$D_{i,air} = \frac{10^{-3}T^{1.75}(\frac{1}{M_i}+\frac{1}{M_{air}})^{0.5}}{P(\sqrt[3]{V_i}+\sqrt[3]{V_{air}})^2}, \tag{2}$$

where M$_{air}$ is the molecular weight of air. In addition, $V_i$ and $V_{air}$ are the corresponding diffusion volumes for molecules $i$ and air. For a molecule, the diffusion volume can be calculated by adding the diffusion volumes of all the atoms. Here we used 15.9 for C, 2.31 for H, and 6.11 for O (Reid et al., 1987).





For the transition regime correction factor $\beta$, we used Fuchs-Sutugin approximation to describe the gas diffusion
in the transition regime (Fuchs and Sutugin, 1971):

$$\beta = \frac{1+Kn}{1+0.3773 \cdot Kn + 1.33 \cdot Kn \cdot (\frac{1+Kn}{\alpha})}, \tag{3}$$

where $Kn$ is the Knudsen number and $\alpha$ is the accommodation coefficient that was set to unity in this study. $Kn$ is
two times the ratio between the mean free path of the molecule $i$ in air ($\lambda_{i,air}$) and particle diameter ($D_p$):

$$Kn = \frac{2\lambda_{i,air}}{D_p}, \tag{4}$$

The value of $\lambda_{i,air}$ can be further expressed following Fuchs and Sutugin (1971):

$$\lambda_{i,air} = \frac{3D_{i,air}}{\overline{c_i}}, \tag{5}$$

where $\overline{c_i}$ is the mean speed of a molecule of $i$ in pure gas of $i$. For a molecule $i$, the value of $\overline{c_i}$ can be described as
follows (Moore, 1962)

$$\overline{c_i} = \sqrt{(\frac{8RT}{\pi M_i})}, \tag{6}$$

The $p_{sat}$ value for each organic compound was estimated using Approximate Bayesian Computation with Sequential
Monte Carlo (ABC-SMC) (Sisson et al., 2007; Toni et al., 2009; Liepe et al., 2014). The ABC-SMC algorithm for
a single parameter (i.e., $p_{sat}$) works by first drawing samples of $p_{sat}$ from a pre-determined prior distribution. For
each sample of $p_{sat}$, a loss function is used to measure the difference between the simulation based on Eq. (1) and
observed evaporation data. Only those samples which have their computed losses below the acceptance threshold
($\varepsilon_1$) will be accepted, resulting in the first posterior distribution of sample size $N$. Subsequently, new samples are
drawn from the previous posterior distribution with a probability proportional to a weight and then are perturbed
as in Toni et al. (2009). Applying the defined loss function and a smaller acceptance threshold ($\varepsilon_2 < \varepsilon_1$), we produce
a new posterior distribution of $N$ samples of $p_{sat}$. The procedure of generating a new set of $N$ accepted samples from
the previous posterior distribution is repeated for $J$ times with always decreasing acceptance thresholds ($\varepsilon_J < \varepsilon_{J-1}$
$< \dots < \varepsilon_2 < \varepsilon_1$). This finally leads to an estimate which can be approximated as the true $p_{sat}$ of the compound of
interest given the observed particle evaporation data.

The ABC-SMC sampling process was performed using the Python package pyABC (Klinger et al., 2018). For each
organic compound, the mean value of $p_{sat}$ and the 95% credible interval (CrI) were calculated using all accepted
samples from the final set of $N$ samples at the end of the pyABC run. The sample size $N$ was defined as 500, with
iteration time $J$ set to 10. The prior distribution was set to be a log-uniform distribution between $10^{-10}$ and $10^{-3}$ Pa.

Sum of the squared residuals between the observed and simulated evaporation data was defined as the loss function.
The number of data points with residence time longer than one hour was typically less than the number of data
points with residence time of one hour or less. To prevent data points with short residence times from dominating
the fittings, the squared difference for each data point with residence time longer than one hour was scaled with a
scaling factor that was the number of data points with residence times of one hour or less. The minimum acceptance





threshold was defined as the total sum of the maximum uncertainties (±1.875% in particle size) from all observation data points. By default, the acceptance thresholds (ε₁ to εⱼ) are automatically calibrated and updated in pyABC. The sampling in pyABC was terminated once either the minimum acceptance threshold or maximum number of iterations (i.e., here set to 10) was reached, whichever came first.

**2.3. Quantum chemical calculations using COSMO*therm***

Conductor-like screening model for real solvents (COSMO-RS; Klamt (1995); Klamt et al. (1998); Eckert and Klamt (2002)) uses a combination of quantum chemistry and statistical thermodynamics to estimate condensed-phase thermodynamic properties, e.g., $p_{sat}$ of pure compounds. The COSMO-RS model is implemented in the COSMO*therm*2021 program (BIOVIA COSMO*therm*, 2021), where the model has been parameterized using experiments of a large set of compounds. The COSMO*therm*2021 program was used to calculate the $p_{sat}$ of the organic compounds. The input files for mono- and dicarboxylic acids and alcohols were taken from the COSMO*base*2021, which contains COSMO*therm* input files of most commonly used compounds. Input files for the other compounds (i.e., DEHS, PEGs) were generated using the BP-TZVPD-FINE-COSMO+GAS_18 template of COSMO*conf*2021 (BIOVIA COSMO*conf*, 2021), which has been created for finding suitable conformer sets for COSMO*therm* calculations. The density functional theory calculations were performed using the TURBOMOLE program version 7.4.1 (Turbomole, 2019). In COSMO*therm*, we used the BP_TZVPD_FINE_21 parametrization and 295 K temperature. For each compound, the $p_{sat}$ in the subcooled liquid state ($p_{sat}^l$) is calculated using its free energy values in the gas ($G^g$) and pure liquid phases ($G^l$):

$$p_{sat}{}^l = e^{-\frac{\left(G^l - G^g\right)}{RT}} \tag{7}$$

Some of the studied compounds can be crystalline solid in bulk at the experimental temperature (i.e., 295 K), which needs to be considered in the $p_{sat}$ calculations. COSMO*therm* is able to estimate saturation vapor pressures in the crystalline solid state ($p_{sat}^s$), if experimental melting temperatures and heats of fusion are given as input. The experimental melting temperatures and heats of fusion used in the COSMO*therm* calculations are correspondingly listed in the Tables S3 and S4 in the Supplement. The free energy of fusion ($\Delta G_{fus}$) at the given temperature is calculated from the given experimental values of melting temperature and enthalpy of fusion and added to the free energy of vaporization. For each compound, its $p_{sat}^s$ can be estimated as follows:

$$p_{sat}{}^s = e^{-\frac{\left(G^l - G^g - \Delta G_{fus}\right)}{RT}} \tag{8}$$

**3. Results and discussion**

With the use of the ABC-SMC method to optimize the evaporation model, we determined the $p_{sat}$ values from the data points collected in the isothermal evaporation measurements under dry conditions at 295 K. The optimized $p_{sat}$ values for the studied compounds are summarized in Table 2. In the following sections for each compound group, we also included the $p_{sat}$ values (at 298 K) reported by previous studies for systematic comparisons. Details of each included study can be found in the Tables S6-S12 in the Supplement. Similar to Bilde et al. (2015), we used two different symbols to distinguish $p_{sat}$ measurements assumed to be taken in crystalline solid or liquid states. In



Figures 1e, 2 and 4, filled squares indicate $p_{sat}$ values assumed to be measured in a crystalline solid state, while open circles represent $p_{sat}$ values assumed to be measured in a liquid state.

**Table 2**. Summary of $p_{sat}$ at 295 K for the organic compounds measured in this study.

| Groups | Compounds | $p_{sat}$ [Pa] |
|---|---|---|
| Polyethylene Glycol | PEG 6 | $2.24^{+0.14}_{-0.14} \times 10^{-5}$ |
| | PEG 7 | $1.06^{+0.02}_{-0.02} \times 10^{-6}$ |
| | PEG 8 | $6.51^{+0.22}_{-0.23} \times 10^{-8}$ |
| | PEG 9 | $6.71^{+9.08}_{-3.86} \times 10^{-9}$ |
| Monocarboxylic acid | Palmitic acid | $5.40^{+0.60}_{-0.53} \times 10^{-6}$ |
| | Stearic acid | $2.42^{+0.17}_{-0.16} \times 10^{-7}$ |
| Dicarboxylic acid | Azelaic acid | $7.61^{+2.38}_{-1.76} \times 10^{-6}$ |
| | Sebacic acid | $1.07^{+0.01}_{-0.01} \times 10^{-7}$ |
| Alcohol | meso-Erythritol | $3.75^{+0.44}_{-0.40} \times 10^{-5}$ |
| | Xylitol | $1.71^{+0.10}_{-0.10} \times 10^{-6}$ |
| Ester | DEHS | $7.52^{+0.18}_{-0.19} \times 10^{-7}$ |

### 3.1. Polyethylene glycols (PEGs)

The measured and simulated EFs of the four PEG compounds are shown in Figure 1a – d. For the remaining
compounds, the measured and simulated EFs can be found in Figures S2 – S5 in the Supplement. With increasing monomer units, the PEG particles showed slower evaporation rates and thus lower volatilities, as expected based on previous measurements (Krieger et al., 2018) and in agreement with their increasing maximum desorption temperatures measured by FIGAERO-CIMS (Ylisirniö et al., 2021). For the investigated PEGs, the $p_{sat}$ values at 295 K were estimated to be in the range between $10^{-9}$ and $10^{-4}$ Pa. Bulk PEG6 – 8 were in liquid states at the
experimental temperature which was above or close to their melting points (Table S3). Given the waxy form of PEG9 at the experimental temperature of 295K, PEG9 particles were most likely in an amorphous solid state and, therefore, the $p_{sat}$ value of PEG9 from our study should be close to that of the subcooled liquid (Bilde et al., 2015). Among these four PEGs, the estimated $p_{sat}$ value of PEG9 has the largest relative uncertainty. This is due to the relatively small size changes of PEG9 particles within the experimental time scale of nearly seven hours. Compared
with literature values for PEG6-9 reported by Krieger et al. (2018), our newly derived values agree within a factor of 2 (Figure 1e).

**Figure 1**. Panels (a) – (d): Measured evaporation factors (EFs; circles) as a function of residence time for PEGs (PEG 6 – 9), simulations with the average optimized $p_{sat}$ values (solid green lines) and 95% credible intervals (95%

CrIs; shaded areas in green), and simulated evaporation curves with a set of reference $p_{sat}$ values ($10^{-9}$ to $10^{-3}$ Pa, with one-decade intervals, dashed grey lines). For the measured data points of EF in (a) – (d), the error bars represent the maximum uncertainty of ±1.875% in particle size measurements on y-axis and the minimum and maximum residence times on x-axis. Panel (e): Measured $p_{sat}$ values for PEGs in this study (red) together with the those reported by Krieger et al. (2018) (yellow).

**3.2. Mono- and dicarboxylic acids**

For the mono- and dicarboxylic acids, the measured $p_{sat}$ values from this study and previous studies are depicted in Figure 2a – d. Our results are in the range between $10^{-7}$ and $10^{-5}$ Pa at 295 K. All these four compounds have melting





points higher than the experimental temperature (Table S3) and therefore the $p_{sat}$ values are here assumed to correspond to solid, crystalline phase, although subcooled liquid phase cannot be ruled out. For each of the four carboxylic acids, the $p_{sat}$ value from this study is in the range of values reported by other independent measurements (Figure 2a – d, b). Among all studies compared in Figure 2, Cappa et al. (2007) and Cappa et al. (2008) reported the lowest solid $p_{sat}$ values from their measurements (Figure 2a – d; bottom rows). Different from other studies, they particularly preheated the particle samples for 30 – 60 min at a temperature slightly above the melting points, in order to remove any solvent molecules which could remain in the particles after sample preparation. Prior to the $p_{sat}$ characterization with the temperature-programmed desorption, the preheated samples were cooled down to 273 K to ensure the particle samples to be in crystalline solid states.

It has been suggested that quick drying after atomization of aqueous organic droplets might not be sufficient to remove all solvents out from the particles (Bilde et al., 2015). Cappa et al. (2007) concluded that the retained solvent molecules might: 1) disrupt the crystal structure at the sample surface to allow many carboxylic acid molecules to exist in configurations favoring evaporation; and 2) increase surface areas for evaporation by increasing surface roughness and porosity. These two effects would possibly increase the evaporation rates of the studied carboxylic acid molecules. For instance, the crystalline-solid $p_{sat}$ values of palmitic acid from our study and other three independent measurements (Davies and Malpass (1961); Tao and Mcmurry (1989); Chattopadhyay and Ziemann (2005)), which used atomized samples after quick drying, are much higher than the crystalline-solid $p_{sat}$ values from samples with less likely impact from solvent molecules (Cappa et al., 2008; Yatavelli and Thornton, 2010) but very close to the liquid $p_{sat}$ value which was predicted by Cappa et al. (2008) based on their crystalline-solid $p_{sat}$ value (Figure 2a). Different from palmitic acid, our crystalline-solid $p_{sat}$ value of stearic acid from atomized samples is very close to the crystalline-solid $p_{sat}$ value but not the liquid one reported by Cappa et al. (2008) (Figure 2b). Furthermore, Saleh et al. (2010) found very similar $p_{sat}$ values of azelaic acid between two sample types which were prepared using atomization (thereby involving solvent; labelled with (a)) and homogeneous condensation (involving no solvent; labelled with (b)) (Figure 2c).

To assess whether retention of solvents impacted our measured $p_{sat}$ values, we normalized the crystalline-solid $p_{sat}$ values from other studies to the one obtained in our study for comparison, as shown in Figure 3. Depending on the studied carboxylic acid, the range of the reported $p_{sat}$ values (grey bars) spans from over one order of magnitude to almost four orders of magnitude. We further categorized studies into two groups according to how the particle samples were prepared. Descriptions of sample preparation in different studies can be found in Tables S7-S10 in the Supplement. In the first group we included studies where aerosol particles were generated by atomization followed by quick drying, while in the second group we included studies where additional procedures (e.g., preheating, homogenous nucleation) were applied to eliminate the solvents from the samples. In general, the group potentially impacted by solvent (blue bar) has higher normalized $p_{sat}$ values, compared the group with no impact from solvent (yellow bar). Divisions between the two sample groups are clearly observed in the two dicarboxylic acids but not in the two monocarboxylic acids. Whether the impact of retained solvent molecules on $p_{sat}$ measurements is compound-dependent or not would require further investigations with more organic compounds.

Among the studied carboxylic acids, the span of the measured $p_{sat}$ values for azelaic acid is relatively large, even when only considering those studies with no impact from solvent molecules. As azelaic acid is a dicarboxylic acid





with nine carbon atoms, polymorphism is one potential factor to be considered when comparing the $p_{sat}$ values between different studies. Previous thermal desorption measurements suggested the presence of polymorphism in dicarboxylic acids with odd number of carbon atoms ($\leq$ 9) (Chattopadhyay and Ziemann, 2005; Salo et al., 2010) as bimodal size distribution was observed during particle evaporation at elevated temperatures (313 – 333 K).

However, we did not observe any bimodal distributions of evaporating particles for azelaic acid in the present study that was carried out at 295 K.

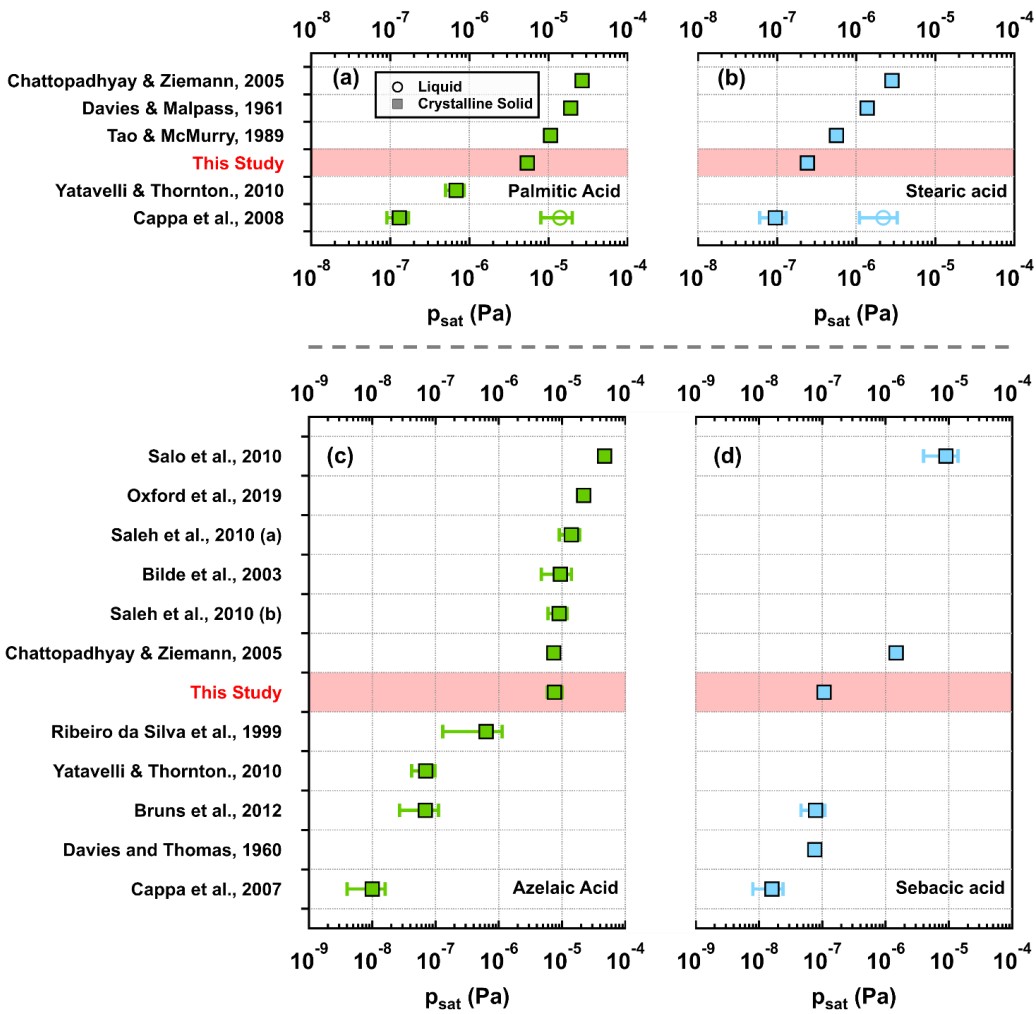

**Figure 2.** Measured $p_{sat}$ values from this study (rectangle in red shaded area) together with those reported in literature for palmitic acid (a; green), stearic acid (b; blue), azelaic acid (c; green), and sebacic acid (d; blue).





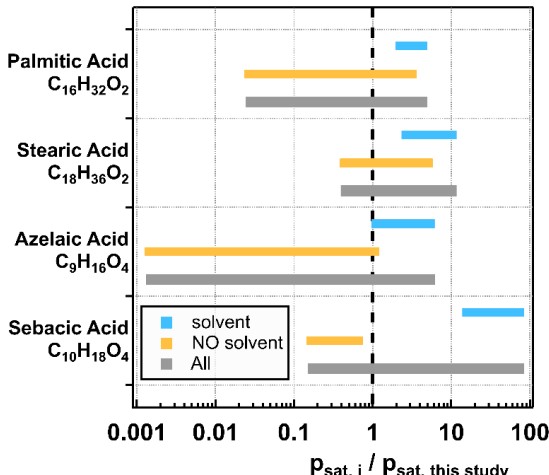


**Figure 3**. Ranges of normalized $p_{sat}$ values for palmitic, stearic, azelaic, and sebacic acids. Each reported $p_{sat}$ value in literature was normalized to that of our study. The whole range of normalized $p_{sat}$ of each compound is shown as a grey bar. The group of samples potentially impacted by solvents is shown as a blue bar, while that with no impact from solvents is shown as a yellow bar.

### 3.3. DEHS and Alcohols

The measured and simulated EFs for DEHS as a function of residence time are presented in Figure S4 in the Supplement. DEHS is in a liquid state at the experimental temperature which is much higher than its melting point (Table S3). The $p_{sat}$ value from our study is $7.52^{+0.18}_{-0.19} \times 10^{-7}$ Pa for DEHS. We could not find any literature $p_{sat}$ value for DEHS. However, dioctyl sebacate (DOS; an isomer with very similar structure) has a comparable $p_{sat}$

value of $2.74 \times 10^{-6}$ Pa at 298 K (Rader et al., 1987).

For xylitol, with an additional -CHOH- in the carbon backbone, the measured $p_{sat}$ value is approximately one order of magnitude lower than that of meso-erythritol (Figure 4; bottom row). The $p_{sat}$ value of meso-erythritol agrees with literature data within less than a factor of 2, while the $p_{sat}$ value of xylitol agrees with that extrapolated from the measurement in Barone et al. (1990) within a factor of 2 (Figure 4).

The melting points of both bulk xylitol and meso-erythritol are above the experimental temperature (Table S3). However, Emanuelsson et al. (2016) observed a bimodal distribution of meso-erythritol particles after evaporation, indicative of two types of particles with different volatilities, and suggested the coexistence of crystalline solid and liquid phases. The presence of such bimodal behavior became increasingly important with the decreasing particle size and/or increasing experimental temperature (298K: < 64 nm; 303 K: < 83 nm; 308 K: < 180 nm) (Emanuelsson

et al., 2016). Cheng et al. (2015) found that aerosol particles smaller than certain critical sizes tend to remain in a liquid state, even at a temperature below the bulk phase transition temperature. However, we did not observe any bimodal distributions of evaporating particles for meso-erythritol in our study. This is likely due to the fact that the particle sizes in our study were mostly larger (61 – 77 nm) than the size range (i.e., < 64 nm at 298 K) which exhibited bimodal evaporation behavior in Emanuelsson et al. (2016). Even though the bimodal behavior of





evaporated meso-erythritol particles was not observed in our study, we cannot rule out the co-existence of crystalline solid and liquid phase states for the meso-erythritol particles.

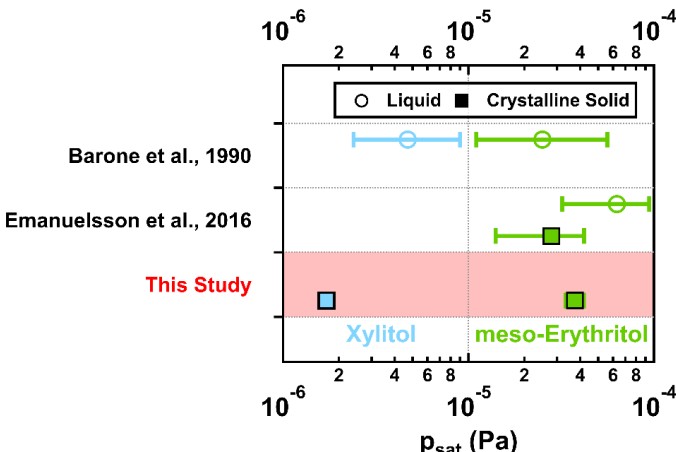

**Figure 4**. Measured $p_{sat}$ values from this study (rectangle in red shaded area) together with those reported in literature for meso-erythritol (green) and xylitol (blue).

**3.4. Comparison with $p_{sat}$ values derived from COSMO*therm* computations**

The $p_{sat}$ values of the organic compounds were estimated using COSMO*therm* calculations. The results for DEHS and PEGs were estimated with the input files generated using the COSMO*conf*2021 program, while those for the studied carboxylic acids and alcohols were computed using the input files from COSMO*base*2021. At the experimental temperature of 295 K, DEHS and the four PEGs (PEG6 – 9) remain in liquid or amorphous states for

their bulk. Thus, only their corresponding $p_{sat}^l$ values were derived from COSMO*therm*. For the remaining compounds of interest (i.e., carboxylic acids, alcohols), the bulk materials are in a crystalline solid state at 295 K. We computed their $p_{sat}^l$ values but also further derived their $p_{sat}^s$ values by accounting for $\Delta G_{fus}$ (see Eq. (8)) for comparing with the measured $p_{sat}$ values. For the $p_{sat}^s$ values, we found the upper and lower limits of estimates using the lowest and highest experimental $\Delta G_{fus}$ values from literature (Table S4), respectively.

We compared the measured $p_{sat}$ values with the $p_{sat}^l$ and/or $p_{sat}^s$ which were estimated by COSMO*therm*, as shown in Figure 5. The $p_{sat}$ values predicted by COSMO*therm* were in reasonable agreement with the experimental values for most of the studied compounds, when considering a reasonable difference of one log unit between measurements and COSMO*therm* estimates. Based on bulk thermodynamics, the carboxylic acids and alcohols would be in a crystalline solid phase at the experimental temperature of 295 K. Their $p_{sat}^s$ values better agree with

the experimental values, compared with the $p_{sat}^l$ values. Exceptions are found in the two dicarboxylic acids for which the $p_{sat}^l$ values (blue open circles) are equally close to or agree better with the experimental $p_{sat}$ value, compared with the $p_{sat}^s$ values (blue filled bars). However, the COSMO*therm* systematically underpredicted the experimental $p_{sat}$ values of PEGs by orders of magnitude. This is the opposite to a previous computational study on multifunctional compounds, where an older parametrization (BP_TZVPD_FINE_18) of COSMO*therm* was seen

to overestimate the $p_{sat}$ by a factor of 5 for every intramolecular hydrogen bond in the compound (Kurtén et al.,





2018). Here, the large underestimation of the $p_{sat}$ of the PEGs by COSMO*therm* suggests that COSMO*therm* overestimates the stability of the condensed phase relative to the gas-phase molecules.

Kurtén et al. (2018) recommended selecting conformers to COSMO*therm* calculations based on their intramolecular hydrogen bonding in order to improve the $p_{sat}$ estimations of multifunctional compounds. The best
agreement between experimental and computed $p_{sat}$ values was found by using only conformers that contain no intramolecular hydrogen bonds in the COSMO*therm* calculations (Kurtén et al., 2018). Here we found that the conformers of multifunctional carboxylic acids and alcohols included in the COSMO*base* produce accurate $p_{sat}$ estimates, even if conformers containing intramolecular hydrogen bonds were not excluded from the calculations. Note that the COSMO*base* conformers of the studied multifunctional carboxylic acids and alcohols were likely
used in the parametrization of the $p_{sat}$ estimates for the quantum chemistry input in the COSMO-RS model. This may explain why the conformers of COSMO*base* produce accurate $p_{sat}$ estimates for these compounds. Systematic conformer sampling and selecting conformer based on their intramolecular hydrogen bonding, suggested by Kurtén et al. (2018), led to higher $p_{sat}$ values (lower $p_{sat}$ for xylitol) than the conformers of COSMO*base*, worsening the agreement with experiments (Table S13). The discrepancy between COSMO*therm*-derived and experimental $p_{sat}$
of the PEGs may be caused by the lack of organic compounds similar to large PEGs in the set of compounds used in the BP_TZVPD_FINE_21 parametrization of COSMO*therm*.

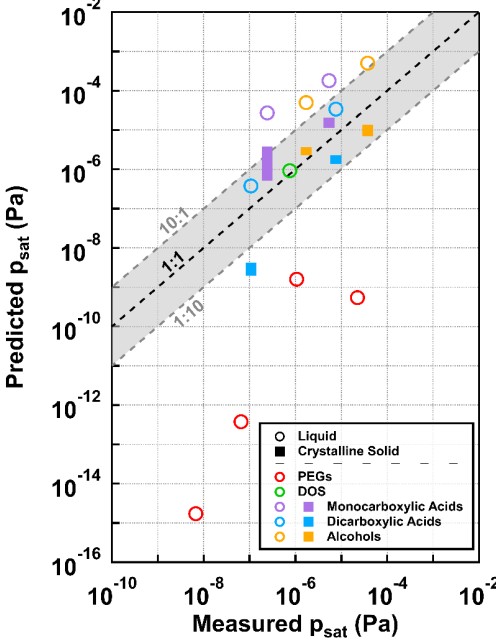

**Figure 5**. Comparison of $p_{sat}$ values between the measurement in this study (x-axis) and different COSMO*therm* predictions (y-axis) at 295 K. The two different markers represent the comparisons of measured $p_{sat}$ values with the
COSMO*therm*-estimated $p_{sat}$ values of liquid (open circles) and crystalline solid (filled bars) phase states, respectively. The height of each filled bar shows the range between the maximum and minimum COSMO*therm*-estimated crystalline solid $p_{sat}$ values. The dashed black line is the 1:1 line, with grey shaded areas showing a deviation of one log unit.





## 4. Conclusions

Here we presented a TDMA-based method to investigate the isothermal evaporation of dry organic nanoparticles that consist of a single compound in a RTC. Using this method, we characterized organic compounds with $p_{sat}$ spanning from $10^{-8}$ to $10^{-4}$ Pa at 295 K. The experimentally determined $p_{sat}$ values from this study are within the ranges of those reported in previous studies based on different measurement techniques. The advantage of our method is that the evaporation measurements are performed at an atmospherically relevant temperature and the

ability to determine $p_{sat}$ values down to $10^{-8}$ Pa in the LVOC range. We acknowledge that there could be uncertainties regarding to the particle phase state and the presence of polymorphism, which have been also discussed in previous volatility studies (Bilde et al., 2003; Chattopadhyay and Ziemann, 2005; Cappa et al., 2007; Emanuelsson et al., 2016). To address such uncertainties would require in-situ spectroscopic methods (e.g., Mie resonance spectroscopy, Raman spectroscopy) that are sensitive enough to differentiate between crystalline solid, amorphous, and subcooled liquid states (Price et al., 2022) and to identify different polymorphic forms (Yeung et

al., 2010). In addition, it has been suggested that aerosol materials with high bulk melting points would be in liquid states at room temperature, once reaching a critical diameter between 12 – 40 nm (Cheng et al., 2015). However, the experimental data on such phase changes are limited to a very small set of compounds (e.g., sodium chloride, ammonium sulfate, polystyrene). Future investigations on the size dependence of phase transitions of LVOCs

should be warranted. Such knowledge will advance our understanding about the phase states but also the volatilities of LVOCs, when they exist in a particle size range (50 – 100 nm) highly relevant to cloud condensation nuclei activation (Kerminen et al., 2012). Furthermore, we found that COSMO*therm* is able to reproduce our measured $p_{sat}$ values for the studied compounds except PEGs, mostly within one order of magnitude. The large difference between the measured and calculated $p_{sat}$ values of PEG highlights the importance of conformer selection in

COSMO*therm* calculations for new types of compounds that have not been used in the parametrization of the model. Unsuitable conformer selection may therefore lead to substantial uncertainties, especially for LVOCs that have not been used in the parameterization of the model.

The current study was focused on $p_{sat}$ characterization for dry, single-compound particles at one set temperature. Nonetheless, our method can be further used to perform isothermal evaporation experiments at different

temperature and RH settings. This will help probe two other important thermodynamics properties - enthalpy of vaporization and organic activity coefficient, which are important to the gas-to-particle partitioning of organic compounds. Furthermore, the method can be extended to study interactions between components in complex particle mixtures, e.g., to study the matrix effect of inorganic salts and non-volatile organics on the isothermal evaporation of LVOCs (Liu et al., 2020).




**Data availability**. The data set is available upon request from the corresponding author.

**Supplement**. The supplement related to this article is available online.

**Author contribution**. ZL designed the study. ZL and MV carried out the laboratory experiments. NH performed the COSMO*therm* calculations. ZL and OPT ran the optimization model. ZL, NH, SS, and TYJ analyzed and interpreted data. ZL wrote the paper with contributions from all coauthors.

**Competing interest.** The authors declare that they have no conflict of interest.

**Acknowledgements**: The authors would like to thank Dr. Angela Buchholz for useful assistance on the measurement setups.

**Financial support.** This research has been supported by the Academy of Finland Flagship program (grant no. 337550), Academy of Finland (grant nos. 317373, 338171, 346371), and the University of Eastern Finland Doctoral Program in Environmental Physics, Health and Biology. We also thank CSC - IT Center for Science, Finland, for computational resources.

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
