# Peer review of "Saturation vapor pressure characterization of selected low-volatility organic compounds using a residence time chamber"

_Atmospheric Chemistry and Physics, 2023_

## Author Response (AR1)

We thank all reviewers for their insightful comments. Below, we provide point-by-point responses to each comment. In the following context, **reviewers' comments and suggestions** are in **black, authors' responses** are in **red**, and **changes to the manuscripts and supplement information** in **blue**. We have also corrected typos and grammatical errors in the manuscript and supplement.

**Reply to Reviewer 1**

The authors present a new setup for measuring the vapor pressures of low-volatility organic compounds. They use an isothermal evaporation method observing the size loss of single-compound nanoparticles under dry conditions at room temperature. In addition, they performed COSMOtherm calculations to predict vapor pressures. They choose a number of compounds to compare to data available in the literature. This comparison is in general quite satisfactory taking the spread of vapor pressures reported previously into account.

The paper is well written, the techniques and data evaluation are well described and the conclusions supported by the data. The topic is relevant for the readers of ACP and I recommend publishing the paper.

I have only one questions/recommendations to the authors. They state on line 167: "… and $\alpha$ is the accommodation coefficient that was set to unity in this study." While I agree that this is a fair assumption because there are no data available, and no evidence for the evaporation coefficient being very small, its value could be easily a factor of 5 smaller, or not? How strong does this uncertainty effect the derived vapor pressure? In addition, the analysis of the data need the surface tension as a parameter. The authors show in their SI the values they took from the literature, but for example for xylitol they simply assume it to be equal to what has been reported for erythritol. All these data carry an uncertainty, which will critically effect the derived vapor pressure. As the authors discuss their retrieval of vapor pressures quite detailed in Section 2.2. I recommend adding a discussion on the propagation of these uncertainties.

Response: Thanks for the insightful comments. Following this suggestion, we have now performed sensitivity analyses on these aspects.

The mass accommodation coefficient ($\alpha$) is commonly assumed to be unity for organics in aerosol studies. It can be either defined as surface accommodation (adsorption, $\alpha_s$) or bulk accommodation (absorption, $\alpha_b$) coefficient (Kolb et al., 2010). In this study, we only focused on single-component systems, and thus no concentration gradient exists between particle bulk and surface for individual particles. In such a case, $\alpha_s$ best describes the evaporation of organic molecules from particle to gas phase. Using in-situ gas- and particle-phase measurements, Krechmer et al. (2017) and Liu et al. (2019) determined $\alpha$ for semi-volatile (SVOCs) or low-volatility organic compounds (LVOCs) across a wide variety of aerosol seeds in chambers. The measured average $\alpha$ spans 0.8 (±0.4) to 1.0 (±0.3), with values close to unity especially for LVOCs. Moreover, molecular dynamics simulations reported that $\alpha_s$ is close to unity (0.96 ~ 1) for SVOCs or LVOCs (Julin et al., 2014; Von Domaros et al., 2020). Therefore, in the context of our study, using $\alpha$ of unity is reasonable, and we estimated a reasonable range of values for a sensitivity test to be from 0.1 to 1.

Regarding the surface tension/energy ($\sigma$), it is unfortunate that there is no literature data for xylitol, However, it is still reasonable to assume its surface tension to be similar to that of meso-xylitol, since these two polyols belong to

the same homologous series. For $\sigma$, we selected $\pm 50\%$ as the range of values for the sensitivity test as an estimate for the uncertainties arising from both the uncertainty of surface tension for particular compounds and the uncertainty due to missing data of the specific compound (xylitol).

To examine their impacts on the estimated $p_{sat}$ values, we performed several sensitivity analyses as shown in Figures R1 – R5 below. As $\alpha$ decreased, the estimated $p_{sat}$ values increased by a factor of two for $\alpha = 0.5$ and increased by a factor of nine for $\alpha = 0.1$, compared with those estimated with $\alpha = 1$. When varying $\sigma$ by $\pm 50\%$, we found mostly small, and in all cases below one order of magnitude, changes in the estimated $p_{sat}$ values for all three tested values of $\alpha$, shown in Table R1. Note that changing $\sigma$ by $\pm 50\%$ led to relatively large impacts on the estimated $p_{sat}$ values for palmitic acid and stearic acid, compared with any other studied compound.

We have added the figures presenting the sensitivity test results in the Figures S2-S6 of Supplementary and provided explanation on how the $\alpha$ and $\sigma$ were chosen in the main text by including following in Section 2.2 and Section 3:

Change: Section 2.2

[…] Chamber partitioning studies (Krechmer et al., 2017; Liu et al., 2019) and molecular dynamic simulations (Julin et al., 2014; Von Domaros et al., 2020) suggested that $\alpha$ was close to unity for LVOCs, and thus $\alpha$ was set to unity in this study. […]

Section 3

Note that there could be uncertainties from the chosen sets of $\alpha$ and $\sigma$ for individual compounds. To examine their impacts on the estimated $p_{sat}$ values, we performed several sensitivity analyses as shown in Figures S2 – S6 of the Supplement. As $\alpha$ decreased, the estimated $p_{sat}$ values increased by a factor of two for $\alpha = 0.5$ and increased by a factor of nine for $\alpha = 0.1$, compared with those estimated with $\alpha = 1$. When varying $\sigma$ by $\pm 50\%$, we found mostly small, and in all cases below one order of magnitude, changes in the estimated $p_{sat}$ values for all three tested values of $\alpha$. Previous experimental (Krechmer et al., 2017; Liu et al., 2019) and computational works (Julin et al., 2014; Von Domaros et al., 2020) have suggested that $\alpha$ is close to unity for the LVOC category, which the selected compounds of this study fall into. Therefore, the optimized $p_{sat}$ values were estimated using $\alpha = 1$.

Table R1. Relative changes in the estimated $p_{sat}$ values upon changes in surface tension ($\sigma$) with varying accommodation coefficients ($\alpha$).

| PEG6 | PEG7 | PEG8 | PEG9 | PMA | STA | AZA | SBA | ERT | XYT | DOS | $\alpha$ | $\sigma$ |
|---|---|---|---|---|---|---|---|---|---|---|---|---|
| | | | | | | | | | | | 1 | Base |
| 15% | 18% | 16% | 41% | 54% | 71% | 38% | 21% | 16% | 21% | 21% | 1 | -50% |
| -13% | -15% | -14% | 10% | -35% | -40% | -25% | -17% | -14% | -17% | -17% | 1 | 50% |
| | | | | | | | | | | | 0.5 | Base |
| 15% | 18% | 16% | 58% | 55% | 71% | 37% | 21% | 17% | 21% | 21% | 0.5 | -50% |
| -13% | -15% | -13% | -12% | -34% | -41% | -23% | -17% | -13% | -17% | -17% | 0.5 | 50% |
| | | | | | | | | | | | 0.1 | Base |
| 15% | 18% | 16% | 37% | 56% | 71% | 35% | 21% | 17% | 21% | 21% | 0.1 | -50% |
| -13% | -15% | -13% | -22% | -33% | -42% | -26% | -17% | -14% | -17% | -17% | 0.1 | 50% |

Note: the base case of $\sigma$ is the refence condition, representing those $\sigma$ values that were used to estimate the $p_{sat}$ values presented in the main text.

[Figure]

**Figure R1**. Sensitivity analyses with different combinations of accommodation coefficient ($\alpha$) (1.0, red; 0.5, green; 0.1, blue) and surface tension/energy ($\sigma$) (-50%, Base, +50%) for PEGs (PEG 6 – 9). The base case of $\sigma$ is the refence condition, representing those $\sigma$ values that were used to determine the $p_{sat}$ values presented in Fig. 1e. The error bars stand for the 95% credible intervals.

[Figure]

**Figure R2**. Sensitivity analyses with different combinations accommodation coefficient ($\alpha$) (1.0, red; 0.5, green; 0.1, blue) and surface tension/energy ($\sigma$) for palmitic acid (PMA, a) and stearic acid (STA, b). The base case of $\sigma$ is the refence condition, representing those $\sigma$ values that were used to determine the $p_{sat}$ values presented in Fig. 2a, b. The error bars stand for the 95% credible intervals.

[Figure]

**Figure R3**. Sensitivity analyses with different combinations of accommodation coefficient ($\alpha$) (1.0, red; 0.5, green; 0.1, blue) and surface tension/energy ($\sigma$) (-50%, Base, +50%) for azelaic acid (AZA, a) and sebacic acid (SBA, b). The base case of $\sigma$ is the refence condition, representing those $\sigma$ values that were used to determine the $p_{sat}$ values presented in Fig. 2c, d. The error bars stand for the 95% credible intervals.

[Figure]

**Figure R4**. Sensitivity analyses with different combinations of accommodation coefficient ($\alpha$) (1.0, red; 0.5, green; 0.1, blue) and surface tension/energy ($\sigma$) (-50%, Base, +50%) for di-2-ethylhexyl sebacate (DEHS). The base case of $\sigma$ is the refence condition, representing those $\sigma$ values that were used to determine the $p_{sat}$ values presented in Section 3.3 in the main text. The error bars stand for the 95% credible intervals.

[Figure]

**Figure R5**. Sensitivity analyses with different combinations accommodation coefficient ($\alpha$) (1.0, red; 0.5, green; 0.1, blue) and surface tension/energy ($\sigma$) for meso-erythritol (ERT, a) and xylitol (XYT, b). The base case of $\sigma$ is the refence condition, representing those $\sigma$ values that were used to determine the $p_{sat}$ values presented in Fig. 4. The error bars stand for the 95% credible intervals.

**Technical comment:**

For the dicaboxylic acids, Bilde et al. (2015) came up with a recommended value and uncertainty range based on all literature available at this time. Maybe add these numbers to your Fig. 2?

Response: This is a good point. We have added these values to Fig.2 and revised the main text by including the following in the section 3.2 and the captions of Figs. 2 and 3.

Change:

[Figure]

**Figure 2.** Measured $p_{sat}$ values from this study (rectangle in red shaded area) together with those reported in literature for palmitic acid (a; green), stearic acid (b; blue), azelaic acid (c; green), and sebacic acid (d; blue). Note that the $p_{sat}$ values and their uncertainties from Bilde et al. (2015) are based on the combined data sets of different studies but not from a particular study or experimental method.

Section 3.2:

[…] To assess whether retention of solvents impacted our measured $p_{sat}$ values, we normalized the crystalline-solid $p_{sat}$ values from other studies (excl. Bilde et al. (2015)) […]

Caption of Figure 3.

Figure 3. […] Note that data from Bilde et al. (2015) shown in Fig. 2 were not included for these analyses.

References sometimes list all authors, sometimes only a few, e.g. Bilde et al., Chemical Reviews 2015.

Response: We updated the author lists in the references.

**Reply to Reviewer 2**

This work discusses a technique to determine saturation vapor pressures using a residence time chamber and monitoring the size change of particles during evaporation. This technique allows for the determination of saturation vapor pressures at atmospherically relevant temperatures and shows promise in determining $p_{sat}$ for low volatility species. In this work single species, dry particles are investigated. This paper was very detailed and provided a lot of great discussion into the challenges of measuring saturation vapor pressures. The writing was clear and the limitations of the study were clearly discussed. The results of this work are reasonable, put into context, and fit well within the scope of ACP. Therefore I would recommend publication after a few minor comments are addressed.

1. I think more discussion on how this could be extend to multicomponent systems would help provide broader implications for this work. It seems like this technique was used in this way in previous work so why was that not discussed at all here?

   Response: Thank you for the suggestion. We have revised the manuscript by providing more discussion regarding the SOA particles at the end of the conclusions section.

   Change: Section 4

   […] Furthermore, the method can be extended to study interactions between compounds in multicomponent aerosol particles, e.g., to study the matrix effect of inorganic salts and non-volatile organics on the isothermal evaporation of LVOCs (Liu et al., 2020). For atmospheric SOA particles which consist of hundreds or even thousands of compounds, their evaporation rates are regulated by the complex interplay between volatility distribution, particle viscosity and particle-phase chemistry. Previous studies with a similar experimental setup have shown biogenic SOA particle evaporation to be dependent on RH (Yli-Juuti et al., 2017), temperature (Li et al., 2019), oxidation levels (Buchholz et al., 2019), or molecular composition (Li et al., 2021; Li et al., 2023). This study, together with the previous studies on SOA particles, have shown the applicability of combining RTC experiments with process modelling. While the RTC method was here used by only probing the size changes during particle evaporation, obtaining molecular-level insights into SOA particle evaporation processes additionally require detailed composition analysis.

2. The introduction discussing the various techniques for measuring $p_{sat}$ was excellent in putting this work into the context of previous work. Including a brief discussion of Cain et al (2020) seems useful though. Some of the novelty here is in being able to do these measurements at atmospherically relevant temperatures and capturing low volatility species, both of which were goals in Cain et al's work. Therefore comparing and contrasting to the technique used in that work could help strengthen the context already provided. (Kerrigan P. Cain, Eleni Karnezi & Spyros N.

Pandis (2020) Challenges in determining atmospheric organic aerosol volatility distributions using thermal evaporation techniques, Aerosol Science and Technology, 54:8, 941-957, DOI: 1080/02786826.2020.1748172)

Response: Thank you for the suggestion. We have now provided a brief discussion of the work from Cain et al (2020) in the introduction and added a couple of sentences to enhance the readability.

Change: Section 1

[…] Without any additional heating, the method can determine $p_{sat}$ values of LVOCs down to $10^{-8}$ Pa by monitoring the evaporation of monodisperse nanoparticles in a residence time chamber (RTC) under dry conditions at room temperature (i.e., 295 K) within timescales of hours. No calibrant with known $p_{sat}$ is required. The RTC method has been also used to study volatilities of biogenic SOA particles in previous studies (Yli-Juuti et al., 2017; Buchholz et al., 2019; Li et al., 2019; Li et al., 2021; Li et al., 2023). Different from our RTC approach, Cain et al. (2020) used a dilution chamber filled with clean air to isothermally dilute the polydisperse aerosol particles by a factor of 10 to initiate particle evaporation. Using the approach of Cain et al. (2020) to estimate particle volatility requires corrections for size-dependent particle wall-loss and coagulation.

Here we used the RTC method to study a set of LVOCs of which $p_{sat}$ values have been reported in literature. […]

3. Equation 2: P is not defined.

Response: Thanks for pointing this out. We added the definition of P as follows:

Change: Section 2.2

where $p$ is the experimental pressure (i.e., 1 atm) and $M_{air}$ is the molecular weight of air.

4. Line 167: Is there a basis in the literature for setting the accommodation coefficient to 1?

Response: Please see also our response to first question by Reviewer 1.

To determine mass accommodation coefficient ($\alpha$) for organic compounds, increasing amounts of experimental and computational studies have been carried out in the past few years. Using in-situ gas- and particle-phase measurements, Krechmer et al. (2017) and Liu et al. (2019) determined $\alpha$ for semi-volatile (SVOCs) or low-volatility organic compounds (LVOCs) across a wide variety of aerosol seeds in chambers. The measured average $\alpha$ spans 0.8 ($\pm$0.4) to 1.0 ($\pm$0.3), with values close to unity especially for LVOCs. Moreover, molecular dynamics simulations reported that $\alpha_s$ is close to unity (0.96 ~ 1) for SVOCs or LVOCs (Julin et al., 2014; Von Domaros et al., 2020). Therefore, in the context of our study, using $\alpha$ of unity is reasonable.

In the revised manuscript we have included additional reasoning and references to support the choice of unity accommodation coefficient as outlined below. We have also included sensitivity test on the mass accommodation coefficient, following the suggestion by Reviewer 1.

Change: Section 2.2

[…] Chamber partitioning studies (Krechmer et al., 2017; Liu et al., 2019) and molecular dynamic simulations (Julin et al., 2014; Von Domaros et al., 2020) suggested that $\alpha$ was close to unity for LVOCs, and thus $\alpha$ was set to unity in this study. […]

5. Line 190: How was an appropriate number of iterations, J, determined?

Response: The number of iterations in ABC-SMC is typically set to 10 to 30 (Schälte and Hasenauer, 2020). For all the studied compounds except sebacic acid, the algorithm was able to converge to the final solution meeting the desired minimum acceptance threshold, with number of iterations less than 10. We have now provided the following sentence for clarification.

Change: Section 2.2

[…] The number of iterations typically ranges from 10 to 30 (Schälte and Hasenauer, 2020), and here we set the iteration time $J$ to 10. […]

6. Figure 3: A minor point, but the caption "no solvent" seems misleading as solvent was used in these experiments. Perhaps "no solvent effects."

Response: We corrected the caption in Figure 3.

**References**

[revised manuscript text omitted]

---

## Author Response (AR2)

We thank the editor for his suggestions. Below, we provide point-by-point responses to each comment. In the following context, **reviewers' comments and suggestions** are in **black, authors' responses** are in red, and changes to the manuscripts and supplement information in blue. We have also corrected typos and grammatical errors in the manuscript and supplement.

**Reply to the editor:**

The study by Li and co-workers is well-written and a substantial contribution to scientific progress in the field. Reviewer comments were addressed adequately.

I would like to follow up on the comments of both reviewers regarding the influence of improper knowledge of accommodation coefficient and surface tension. The authors now carefully discuss the sensitivities and it seems that this parametric uncertainty in some cases significantly exceeds the (sometimes quite small) error coming from determination of dDp/dt. Hence, I suggest not only discussing the uncertainty, but also propagating an estimate of these parametric uncertainties into the reported uncertainty in p_sat.

Response: Thanks for the suggestion. We now use the $p_{sat}$ ranges that were estimated using $\alpha = 1$ and $\pm$ 50% variation in the chosen $\sigma$ as the error bars for the measured $p_{sat}$ values. This means that we have included the effect of uncertainty of surface tension to our uncertainty estimates of $p_{sat}$, but all values of $p_{sat}$ reported in the main text were estimated using $\alpha = 1$. As reasoned in our reply to reviewers and in the second paragraph of Section 3 in the revised manuscript, we consider $\alpha = 1$ to be a reasonable choice and therefore have not accounted for other values of $\alpha$ in the uncertainty estimate of $p_{sat}$.

We have revised the error bars of $p_{sat}$ values for this study in Table 2, and Figures 1e, 2, 4, and 5 accordingly. Now the presented uncertainty ranges for $p_{sat}$ include $\pm$ 50% variation in the chosen $\sigma$ and the fitting uncertainties (i.e., 95 % CrIs). The contributions of the fitting uncertainties to the presented uncertainty ranges are summarized in Table R1. The revised figures are also shown here below. We have also revised the text in Section 3 in the revised manuscript to explain the error bars.

Table R1. Relative contributions of the fitting uncertainties (95% credible intervals) to the presented error bars.

| Cases of $\sigma$ | PEG6 | PEG7 | PEG8 | PEG9 | PMA | STA | AZA | SBA | ERT | XYT | DOS |
|---|---|---|---|---|---|---|---|---|---|---|---|
| Base | 32% | 9% | 19% | 95% | 18% | 11% | 50% | 4% | 43% | 23% | 11% |
| -50% | 36% | 10% | 19% | 80% | 32% | 16% | 50% | 4% | 52% | 28% | 15% |
| 50% | 22% | 8% | 15% | 51% | 14% | 42% | 35% | 3% | 30% | 18% | 10% |

Change:

Section 3

[revised manuscript text omitted]

Technical Comments:

l. 20 - I suggest indicating that DEHS is an ester in analogy to the other listed compounds

Response: We modified the sentence in the introduction.

Change: Introduction

[…] The compounds included four polyethylene glycols (PEG: PEG6, PEG7, PEG8 and PEG9), two monocarboxylic acids (palmitic acid and stearic acid), two dicarboxylic acids (azelaic acid and sebacic acid), two alcohols (meso-erythritol and xylitol), and one ester (di-2-ethylhexyl sebacate). […]

l. 176 - should read "molecular dynamics simulations"

Response: We corrected the typo.

Figure 3 - I suggest to decapitalize "NO" in legend to avoid confusion with nitrogen monoxide

Response: Now we use the "No solvent effect" in the legend.

Change: